# GraphNVP: An Invertible Flow-based Model for Generating Molecular Graphs

## Abstract

We propose GraphNVP, an invertible flow-based molecular graph generation model. Existing flow-based models only handle node attributes of a graph with invertible maps. In contrast, our model is the first invertible model for the *whole graph components*: both dequantized node attributes and adjacency tensor are converted into latent vectors through two novel invertible flows. This decomposition yields the exact likelihood maximization on graph-structured data. We decompose the generation of a graph into two steps: generation of (i) an adjacency tensor and (ii) node attributes. We empirically demonstrate that our model and the two-step generation efficiently generates valid molecular graphs with almost no duplicated molecules, although there are no domain-specific heuristics ingrained in the model. We also confirm that the sampling (generation) of graphs is faster in order of magnitude than other models in our implementation. In addition, we observe that the learned latent space can be used to generate molecules with desired chemical properties. Finally we list open problems for this new direction of fully invertible graph generation researches.

## 1 Introduction

Generation of molecules with certain desirable properties is a crucial problem in computational drug discovery. Recently, deep learning approaches are being actively studied for generating promising candidate molecules quickly. Earlier models (Kusner et al., 2017; Gómez-Bombarelli et al., 2018) depend on a string-based representation of molecules. However, recent models (Jin et al., 2018; You et al., 2018a; De Cao & Kipf, 2018) directly work on molecular graph representations and record impressive experimental results. In these studies, either variational autoencoder (VAE) (Kingma & Welling, 2014) or generative adversarial network (GAN) (Goodfellow et al., 2014; Radford et al., 2015) are used mainly to learn mappings between the graphs and their latent vector representations.

In this paper, we propose *GraphNVP*, yet another framework for molecular graph generation based on the invertible normalizing flow, which was mainly adopted for image generation tasks (Dinh et al., 2017; Kingma & Dhariwal, 2018). To capture distributions of irregular graph structure of molecules into a latent representation, we propose a novel two-step generation scheme. Specifically, GraphNVP is equipped with two latent representations for a molecular graph: first for the graph structure represented by an adjacency tensor, and second for node (atom) attributes. We introduce two types of reversible flows that work for the aforementioned two latent representations of graphs.

Recent work by Liu et al. (2019) proposes a flow-based invertible model for transforming the node attribute matrix. However, they use a non-invertible encoder for transforming the adjacency tensor making the complete model non-invertible. Our model is the first fully invertible model for the *whole graph components*: both adjacency tensor and node attributes are converted into latent vectors through two novel invertible flows.

To sample a graph, we develop a novel two-step generation process. During the generation process, GraphNVP first generates the graph structure. Then node attributes are generated according to this structure. This two-step generation enables efficient generation of valid molecular graphs. The full reversibility of our model on graphs contributes to two major benefits: a simple architecture and precise log-likelihood maximization. A major advantage of invertible models is that we do not need to design a separate decoder for sample generation: new graph samples can be generated by simply feeding a latent vector into the same model but in the reverse order.

In contrast, VAE models require an encoder and a separated decoder. Decoding processes of several VAE graph generators are often quite complicated to assure valid generations (Kusner et al., 2017; Jin et al., 2018; Ma et al., 2018), and computing a graph reconstruction loss may require expensive graph matching (Simonovsky & Komodakis, 2018). The lack of an encoder in GAN models makes it challenging to manipulate the sample generation. For example, it is not straightforward to use a GAN model to generate graph samples that are similar to a query graph (e.g., lead optimization for drug discovery), while it is easy for flow-based models.

Unlike VAEs and GANs, invertible models are capable of precise log-likelihood evaluation. We believe precise optimization is crucial in molecule generation for drugs, which are highly sensitive to a minor replacement of a single atom (node).

In the experiments, we compare the proposed flow model with several existing graph generation models using two popular molecular datasets. The proposed flow model generates molecular graphs with almost 100% uniqueness ratio: namely, the results contain almost no duplicated molecular graphs without ingrained domain expert knowledge and extra validity checks. The proposed model enjoys fast graph samplings; faster in orders of magnitude than other graph generation models in our implementation. Additionally, we show that the learned latent space can be utilized to generate molecular graphs with desired chemical properties, even though we do not encode domain expert knowledge into the model. Finally we list open problems for the development of this new direction of fully invertible graph generation researches.

## 2 RELATED WORK

### 2.1 MOLECULAR GRAPH GENERATION

We can classify the existing molecular graph generation models based on how the data distribution is learned. Most current models belong to two categories. First, VAE-based models assume a simple variational distribution for latent representation vectors (Jin et al., 2018; Liu et al., 2018; Ma et al., 2018). Second, some models implicitly learn the empirical distribution, especially based on the GAN architecture (e.g., (De Cao & Kipf, 2018; You et al., 2018a; Guimaraes et al., 2017)). Some may resort to reinforcement learning (You et al., 2018a) to alleviate the difficulty of direct optimization of the objective function. We also observe an application of autoregressive recurrent neural networks (RNN) for graphs (You et al., 2018b). In this paper, we will add a new category to this list: namely, the invertible flow.

Additionally, we can classify the existing models based on the process they use for generating a graph. There are mainly two choices in the generation process. One is a sequential *iterative* process, which generates a molecule in a step-by-step fashion by adding nodes and edges one by one (Jin et al., 2018; You et al., 2018a). The alternative is *one-shot* generation of molecular graphs, when the graph is generated in a single step. This process resembles commonly used image generation models (e.g., (Kingma & Dhariwal, 2018)). The former process is advantageous in (i) dealing with large molecules and (ii) forcing validity constraints on the graph (e.g., a valency condition of molecule atoms). The latter approach has a major advantage: the model is simple to formulate and implement. This is because the one-shot approach does not have to consider arbitrary permutations of the sequential steps, which can grow exponentially with the number of nodes in the graph.

Combining these two types of classification, we summarize the current status of molecular graph generation in Table 1. In this paper, we propose the first graph generation model based on the invertible flow, with one-shot generation strategy.

### 2.2 INVERTIBLE FLOW MODELS

To the best of our knowledge, the invertible flow was first introduced to the machine learning community by (Tabak & Vanden-Eijnden, 2010; Tabak & Turner, 2013). Later, Rezende & Mohamed (2015) and Dinh et al. (2015) leveraged deep neural networks in defining tractable invertible flows. Dinh et al. (2015) introduced reversible transformations for which the log-determinant calculation is tractable. These transformations, known as *coupling layers*, serve as the basis of recent flow-based image generation models (Dinh et al., 2017; Kingma & Dhariwal, 2018; Grathwohl et al., 2019)

| Name | Distribution Model | | | | | Generation Process | |
|---|---|---|---|---|---|---|---|
| | VAE | Adversarial | RL | RNN | Inv.Flow | Iterative | OneShot |
| RVAE (Ma et al., 2018) | ✓ | - | - | - | - | - | ✓ |
| CGVAE (Liu et al., 2018) | ✓ | - | - | - | - | ✓ | - |
| JT-VAE (Jin et al., 2018) | ✓ | - | - | - | - | ✓ | - |
| MolGAN (De Cao & Kipf, 2018) | - | ✓ | - | - | - | - | ✓ |
| GCPN (You et al., 2018a) | - | ✓ | ✓ | - | - | ✓ | - |
| GraphRNN (You et al., 2018b) | - | - | - | ✓ | - | ✓ | - |
| **GraphNVP** | - | - | - | - | ✓ | - | ✓ |

Table 1: Existing models of molecular graph generation. We propose the first invertible flow-based graph generation model in the literature..

Readers are referred to the latest survey (Kobyzev et al., 2019) for the general flow methodologies.

So far, the application of flow-based models is mostly limited to the image domain. As a few exceptions, Kumar et al. (2018) proposed flow-based invertible transformations on graphs. However, their model is only capable of modeling the node assignments and cannot learn a latent representation of the adjacency tensor; therefore, it cannot generate a graph structure. Liu et al. (2019) proposed to plug a non-invertible decoder for the adjacency tensor to this flow model afterwards, giving up training the entire graph generator in a single unified estimator. We overcome this issue by introducing two latent representations, one for node assignments and another for the adjacency tensor, to capture the unknown distributions of the graph structure and its node assignments. Thus, we consider our proposed model to be the first invertible flow-based model that can generate attributed graphs including the adjacency structure.

## 3 GRAPHNVP: FLOW-BASED GRAPH GENERATION MODEL

### 3.1 FORMULATION

We use the notation $G = (A, X)$ to represent a graph $G$ consisting of an adjacency tensor $A$ and a feature matrix $X$. Let there be $N$ nodes in the graph. Let $M$ be the number of types of nodes and $R$ be the number of types of edges. Then $A \in \{0, 1\}^{N \times N \times R}$ and $X \in \{0, 1\}^{N \times M}$. In the case of molecular graphs, $G = (A, X)$ represents a molecule with $R$ types of bonds (single, double, etc.) and $M$ types of atoms (e.g., oxygen, carbon, etc.). Our objective is to learn an invertible model $f_\theta$ with parameters $\theta$ that maps $G$ into a latent point $z = f_\theta(G) \in \mathbb{R}^{D=(N \times N \times R)+(N \times M)}$. We describe $f_\theta$ as a normalizing flow composed of multiple invertible functions.

Let $z$ be a latent vector drawn from a known prior distribution $p_z(z)$ (e.g., Gaussian): $z \sim p_z(z)$. With the change of variable formula, the log probability of a given graph $G$ can be calculated as:

$$\log (p_G(G)) = \log (p_z(z)) + \log \left( \left| \det \left( \frac{\partial z}{\partial G} \right) \right| \right), \tag{1}$$

where $\frac{\partial z}{\partial G}$ is the Jacobian of $f_\theta$ at $G$.

### 3.2 GRAPH REPRESENTATION

Directly applying a continuous density model on discrete components may result in degenerate probability distributions. Therefore, we cannot directly employ the change of variable formula (Eq. 1) for these components. The same issue, especially modeling the discrete structure of the adjacency $A$, has been a problem in existing one-shot generators based on GAN (De Cao & Kipf, 2018) and VAE (Ma et al., 2018). They resort to an ad-hoc workaround; treating the adjacency tensor as a real-valued continuous tensor. In this paper we take another approach, *dequantization* (Theis et al., 2016), following the flow-based image generation models (Dinh et al., 2017; Kingma & Dhariwal, 2018). The dequantization process adds uniform noises to $A$ and $X$ and yield the dequantized graph component $G' = (A', X')$. Specifically, $A' = A + cu$; $u \sim U[0, 1)^{N \times N \times R}$ and $X' = X + cu$; $u \sim U[0, 1)^{N \times M}$, where $0 < c < 1$ is a scaling hyperparameter ($c = 0.9$ is adopted for our experiment).

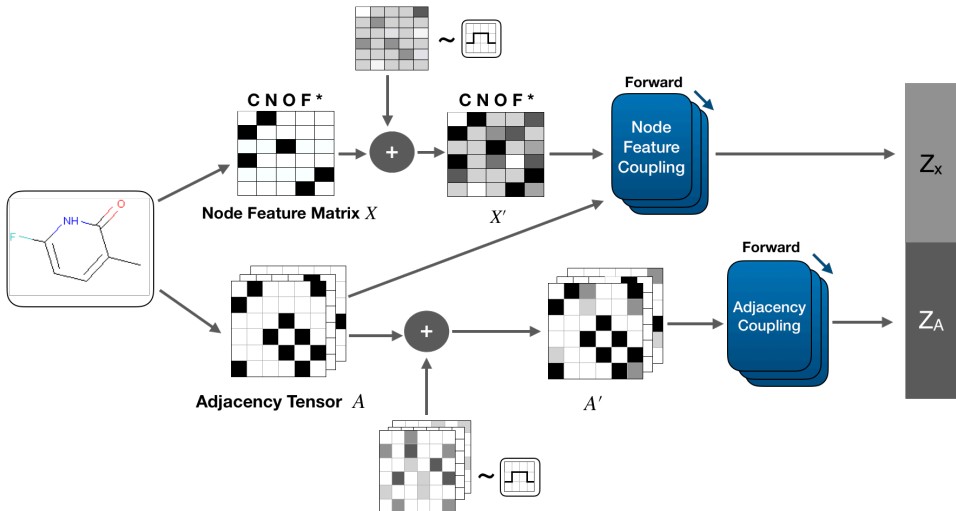

Figure 1: Forward transformation of the proposed GraphNVP. The original discrete $A$ and $X$ are first dequantized into $A'$ and $X'$. Layers of invertible couplings are applied to these dequantized tensors.

This $G'$ is used as the input in Eq. 1. Note that the original discrete inputs $A$ and $X$ can be recovered by *quantization*: simply applying floor operation on each continuous value in $A'$ and $X'$.

Hereafter, all the transformations consisting $f_\theta$ are performed on dequantized inputs $A'$ and $X'$, not on $A$ and $X$. It means $f_\theta$ is a bijective function that maps $G' \to z$: thus $f^{-1}(z)$ returns the dequantized $G'$, not the original $G$. However, our generative model can recover the original discrete $G$ by performing the postprocessing quantization to inverted $G'$.

There are a few works related to discrete invertible flows such as (Hoogeboom et al., 2019; Tran et al., 2019). The former maps discrete data $x$ to a discrete latent space. However, we prefer a smoothly distributed continuous latent space for molecule decoration and optimization applications (see Sec. 4.3). The latter can map discrete data $x$ to a continuous $z$, but computation includes approximation. Approximated likelihood evaluations decreases the significance of the invertible flows against VAEs. So we do not adopt these options in this paper.

### 3.3    COUPLING LAYERS

Based on real-valued non-volume preserving (real NVP) transformations introduced in (Dinh et al., 2017), we propose two types of reversible affine coupling layers; *adjacency coupling* layers and *node feature coupling* layers that transform the adjacency tensor $A'$ and the feature matrix $X'$ into latent representations, $z_A \in \mathbb{R}^{N \times N \times R}$ and $z_X \in \mathbb{R}^{N \times M}$, respectively.

We apply $L_X$ layers of node feature coupling layers to a feature matrix $X'$ to obtain $z_X$. We denote an intermediate representation of the feature matrix after applying the $\ell^{\text{th}}$ node feature coupling layer as $z_X^{(\ell)}$. Starting from $z_X^{(0)} = X'$, we repeat updating rows of $z_X$ over $L_X$ layers. Each row of $z_X^{(\ell)}$ corresponds to a feature vector of a node in the graph. Finally, we obtain $z_X = z_X^{(L_X)}$ as the final latent representation of the feature matrix. The $\ell^{\text{th}}$ node feature coupling layer updates a single row $\ell$ of the feature matrix while keeping the rest of the input intact:

$$z_X^{(\ell)}[\ell, :] \leftarrow z_X^{(\ell-1)}[\ell, :] \odot \exp\left(s(z_X^{(\ell-1)}[\ell^-, :], A)\right) + t(z_X^{(\ell-1)}[\ell^-, :], A), \qquad (2)$$

where functions $s$ and $t$ stand for scale and translation operations, and $\odot$ denotes element-wise multiplication. We use $z_X[\ell^-, :]$ to denote a latent representation matrix of $X'$ excluding the $\ell^{\text{th}}$ row (node). Rest of the rows of the feature matrix will stay the same as

$$z_X^{(\ell)}[\ell^-, :] \leftarrow z_X^{(\ell-1)}[\ell^-, :]. \qquad (3)$$

Both $s$ and $t$ can be formulated with arbitrary nonlinear functions, as the reverse step of the model does not require inverting these functions. Therefore, we use the graph adjacency tensor $A$ when

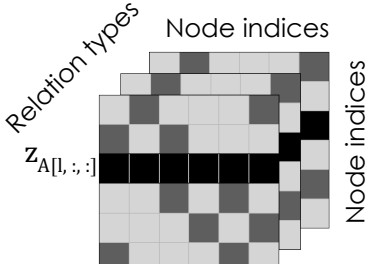
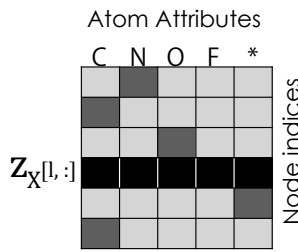

Figure 2: Masking schemes used in proposed affine coupling layers. Left: Adjacency coupling layer: A single row of adjacency tensor is masked. Right: Node feature coupling layer: All channels belonging to a single node are masked.

computing invertible transformations of the node feature matrix $X'$. So as functions $s$ and $t$ in a node feature coupling layer, we use a sequence of generic graph neural networks. It should be noted that we use the discrete adjacency tensor $A$, as only the node feature matrix is updated in this step. In this paper, we use a variant of Relational GCN (Schlichtkrull et al., 2018) architecture.

Likewise, we apply $L_A$ layers of transformations for the adjacency tensor $A'$ to obtain the latent representation $z_A$. We denote an intermediate representation of the adjacency tensor after applying the $\ell^{\text{th}}$ adjacency coupling as $z_A^{(\ell)}$. The $\ell^{\text{th}}$ adjacency coupling layer updates only a single slice of $z_A^\ell$ with dimensions $N \times R$ as:

$$z_A^{(\ell)}[\ell, :, :] \leftarrow z_A^{(\ell-1)}[\ell, :, :] \odot \exp\left(s(z_A^{(\ell-1)}[\ell^-, :, :])\right) + t(z_A^{(\ell-1)}[\ell^-, :, :]). \tag{4}$$

The rest of the rows will stay as it is:

$$z_A^{(\ell)}[\ell^-, :, :] \leftarrow z_A^{(\ell-1)}[\ell^-, :, :]. \tag{5}$$

For the adjacency coupling layer, we adopt multi-layer perceptrons (MLPs) for $s$ and $t$ functions. Starting from $z_A^{(0)} = A'$, we repeat updating the first axis slices of $z_A$ over $L_A$ layers. Finally, we obtain $z_A = z_A^{(L_A)}$ as the final latent representation of the adjacency tensor.

### 3.3.1 MASKING PATTERNS AND PERMUTATION OVER NODES

Eqs. (2, 4) are implemented with masking patterns shown in Figure 2. Based on experimental evidence, we observe that masking $z_A(A')$ and $z_X(X')$ w.r.t. the node axis performs the best. Because a single coupling layer updates one single slice of $z_A$ and $z_X$, we need a sequence of $N$ coupling layers at the minimum, each masking a different node, for each of the adjacency coupling and the node feature coupling layers.

We acknowledge that this choice of masking axis over $z_X$ and $z_A$ makes the transformations not invariant to permutations of the nodes. We can easily formulate permutation-invariant couplings by changing the slice indexing based on the non-node axes (the $3^{\text{rd}}$ axis of the adjacency tensor, and the $2^{\text{nd}}$ axis of the feature matrix). However, using such masking patterns results in dramatically worse performance due to the sparsity of molecular graphs. For example, organic compounds are mostly made of carbon atoms. Thus, masking the carbon column in $X'$ (and $z_X$) results in feeding a nearly-empty matrix to the scale and the translation networks, which is almost non-informative to update the carbon column entries of $X'$ and $z_X$. We consider this permutation dependency as a limitation of the current model, and we intend to work on this issue as future work.

### 3.4 TRAINING

During the training, we perform the forward computations shown in Figure 1 over minibatches of training data ($G = (A, X)$) and obtain latent representations $z = \text{concat}(z_A, z_X)$. Our objective is maximizing the log likelihood $p_G(G)$ (Eq. 1) over minibatches of training data. This is implemented as minimization of the negative log likelihood using the Adam optimizer (Kingma & Ba, 2015).

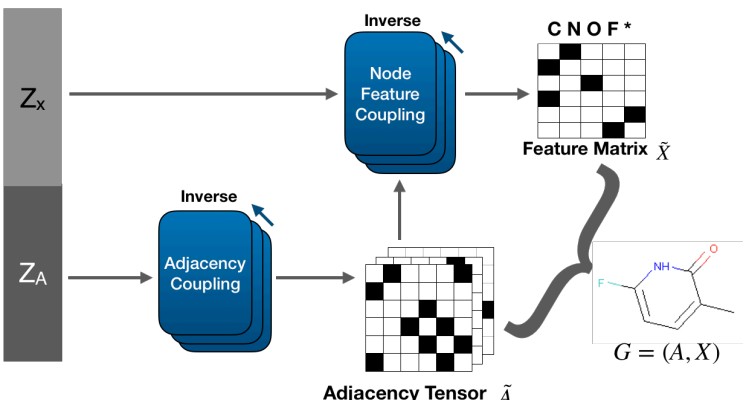

Figure 3: Generative process of the proposed GraphNVP. We apply the inverse of the coupling layers in the reverse order, so that the dequantized inputs $\tilde{A}$ and $\tilde{X}$ are recovered. Additional quantization reconstructs the original discrete graph, $G = (A, X)$.

## 3.5 TWO-STEP MOLECULAR GRAPH GENERATION

Because our proposed model is invertible, graph generation is simply executing the process shown in Figure 1 in reverse. During the training, node feature coupling and adjacency coupling can be performed in either order, as the output of one coupling module does not depend on the output of the other coupling module. However, because the node feature coupling module requires a valid adjacency tensor as an input, we also need an adjacency tensor to perform the reverse step of node feature coupling. Therefore, we apply the reverse step of adjacency coupling module first, so we get an adjacency tensor as the output. Next, the adjacency tensor is fed into the reverse step of the node feature coupling. The generation process is shown in Figure 3. In section 4, we show that this *2-step generation process* can efficiently generate chemically valid molecular graphs.

**1st step:** We draw a random sample $z = \text{concat}(z_A, z_X)$ from prior $p_z$ and split sampled $z$ into $z_A$ and $z_X$. Next, we apply a sequence of *inverted* adjacency coupling layers on $z_A$. As a result, we obtain a probabilistic adjacency tensor $\tilde{A}'$, from which we construct a discrete adjacency tensor $\tilde{A} \in \{0, 1\}^{N \times N \times R}$ by taking node-wise and edge-wise argmax.

**2nd step:** We generate a feature matrix given the sampled $z_X$ and the generated adjacency tensor $\tilde{A}$. We input $\tilde{A}$ along with $z_X$ into a sequence of *inverted* node feature coupling layers to attain $\tilde{X}'$. Likewise, we take node-wise argmax of $\tilde{X}'$ to get discrete feature matrix $\tilde{X} \in \{0, 1\}^{N \times M}$.

## 4 EXPERIMENTS

### 4.1 PROCEDURE

We use two popular chemical molecular datasets, QM9 (Ramakrishnan et al., 2014) and ZINC-250k (Irwin et al., 2012). QM9 dataset contains 134k molecules, and ZINC-250k is made of 250k drug-like molecules randomly selected from the ZINC database. The maximum number of atoms in a molecule are 9 for the QM9 and 38 for the ZINC, respectively (excluding hydrogen). Following a standard procedure, we first kekulize molecules and then remove hydrogen atoms from them. The resulting molecules contain only single, double, and triple bonds.

We convert each molecule to an adjacency tensor $A \in \{0, 1\}^{N \times N \times R}$ and a feature matrix $X \in \{0, 1\}^{N \times M}$. N is the maximum number of atoms a molecule in a certain dataset can have. If a molecule has less than $N$ atoms, we insert virtual nodes as padding to keep the dimensions of $A$ and $X$ the same for all the molecules. Because the original adjacency tensors can be sparse, we add a virtual bond edge between the atoms that do not have a bond in the molecule. Thus, an adjacency tensor consists of $R = 4$ adjacency matrices stacked together, each corresponding to the existence

| Method | QM9 | | | | ZINC | | | |
|---|---|---|---|---|---|---|---|---|
| | % V | % N | % U | % R | % V | % N | % U | % R |
| **GraphNVP** | 83.1 ($\pm$ 0.5) | 58.2 ($\pm$ 1.9) | 99.2 ($\pm$ 0.3) | 100.0 | 42.6 ($\pm$ 1.6) | 100.0 ($\pm$ 0.0) | 94.8 ($\pm$ 0.6) | 100.0 |
| RVAE | 96.6 | 97.5 | - | 61.8 | 34.9 | 100.0 | - | 54.7 |
| MolGAN | 98.1 | 94.2 | 10.4 | - | - | - | - | - |
| GVAE | 60.2 | 80.9 | 9.3 | 96.0 | 7.2 | 100.0 | 9.0 | 53.7 |
| CVAE | 10.3 | 90.0 | 67.5 | 3.6 | 0.7 | 100.0 | 67.5 | 44.6 |
| JT-VAE | - | - | - | - | 100.0 | 100.0 | 100.0 | 76.7 |
| CG-VAE | 100.0 | 94.4 | 98.6 | - | 100.0 | 100.0 | 99.8 | - |

Table 2: Performance of generative models with respect to quality metrics. Baseline scores are borrowed from the original papers. Zinc results of JT-VAE are reproduced based on the settings written in the original paper. Scores of GraphNVP are averages over 5 runs. Standard deviations are presented below the average scores.

of a certain type of bond (single, double, triple, and virtual bonds) between the atoms. The feature matrix is used to represent the type of each atom (e.g., oxygen, fluorine, etc.).

We use a multivariate Gaussian distribution $\mathcal{N}(\mathbf{0}, \sigma^2 \boldsymbol{I})$ as prior distribution $p_z(z)$, where standard deviation $\sigma$ is learned simultaneously during the training. More details are presented in the appendix.

## 4.2 NUMERICAL EVALUATION

Following (Kingma & Dhariwal, 2018), we sample 1,000 latent vectors from a temperature-truncated normal distribution $p_{z,T}(z)$ (see the appendix for details) and transform them into molecular graphs by performing the reverse step of our model. We compare the performance of the proposed model with baseline models in Table 2 using following metrics. **Validity (V)** is the percentage of generated graphs corresponding to valid molecules. **Novelty (N)** is the percentage of generated valid molecules not present in the training set. **Uniqueness (U)** is the percentage of unique valid molecules out of all generated molecules. **Reconstruction accuracy (R)** is the percentage of molecules that can be reconstructed perfectly by the model: namely, the ratio of molecules $G$ s.t. $G = f_\theta^{-1}(f_\theta(G))$.

We choose Regularizing-VAE (RVAE) (Ma et al., 2018) and MolGAN (De Cao & Kipf, 2018) as baseline one-shot generation models. We compare with two additional models: grammar VAE(GVAE) (Kusner et al., 2017) and character VAE (CVAE)(Gómez-Bombarelli et al., 2018), which learn to generate string representations of molecules. Finally, JT-VAE (Jin et al., 2018) and CG-VAE (Ma et al., 2018) as the state-of-the-art iterative generation models with complicated decoders with validity checkers.

Notably, proposed GraphNVP guarantees 100% reconstruction accuracy, attributed to the invertible function construction of normalizing flows. Also, it is notable that GraphNVP enjoys a significantly high uniqueness ratio. Although some baselines exhibit a higher validity on QM9 dataset, the set of generated molecules contains many duplicates. Additionally, we want to emphasize that our model generates a substantial number of valid molecules without explicitly incorporating the chemical knowledge as done in some baselines (e.g., valency checks for chemical graphs in RVAE, MolGAN, JT-VAE, and CG-VAE. This is preferable because additional validity checks consume computational time (see Sec.4.2.1), and may result in a low reconstruction accuracy (e.g., RVAE and JT-VAE). As GraphNVP does not incorporate domain-specific procedures during learning, it can be easily used for learning generative models on general graph structures. Two iterative generation models, JT-VAE (Jin et al., 2018) and CG-VAE (Liu et al., 2018), show great results in the table. However, decoders of these models are quite complicated to properly implement and reproduce the same performance. In contrast, the proposed GraphNVP enjoys a simple network architecture and its decoder is immediately available by just inverting the trained coupling layers.

Considering the simplicity of the model, proposed GraphNVP achieves good performance among latest graph generation models. We guess the generation scheme of the GraphNVP may affect these performances in part. The proposed generation scheme is in the midst of the one-shot and the iterative graph generation. From a higher perspective, our generation is one-shot: once we sample the latent

vector $z = [z_X, z_A]$ then the final output graph is determined. In a detailed observation, the inversion process is iterative: for each (inverting) $\ell$-th layer of two couplings, the network recovers a adjacency matrix or a feature vector of a $\ell$-th node, given representations of all the nodes except $\ell$-th node. One layer of partition-based affine coupling is not a mapping of super-flexible, but may be flexible enough to warp a single node's representation.

### 4.2.1 COMPUTATIONAL TIME FOR GRAPH GENERATION

One practically important aspect of graph generation is computational time. Training and sampling a generative model is much faster than wet-lab experiments, but the computational time is still an issue for tasks involving huge search spaces: e.g. drug search. We compare the computational time (wall-clock time) for sampling $1,000$ graphs for ZINC dataset experiment runs. The average wall-clock time (excluding preprocessing time) of GraphNVP for sampling is only 4.6 [sec] (implemented in Chainer Tokui et al. (2015)). This is faster in order of magnitude than several baselines (in our test runs): 193.5 [sec] for CVAE (Tensorflow), 460 [sec] for GVAE (Tensorflow), and 124 [sec] for JT-VAE (pytorch).

The sampling time affects the number of valid, novel, and unique molecular graphs we can collect within a unit time. The validity of the GraphNVP samples are relatively low, but still keeps $40\%$. In contrast, sampling time is 30 to 100 times shorter. Thus we can obtain more (10 to 40 times) valid, novel, and unique molecules in the same computation time. Once we obtained the generated molecule, we usually calculate or predict the value of specific property in computer to check the generated molecules have desired values. Thus generating many molecules increases the chance to discover molecule with required property. Assume we need to prepare 1 million unique, novel, and valid molecules from models trained via ZINC dataset. With a very rough estimate, we expect the GraphNVP, JT-VAE, and GVAE requires 1.1 hours, 1.5 days, and 121 5 days, respectively. Such slow graph generations would harm the productivity of the R&D projects. Further, this will reduce the usage of cloud computing servers such as Amazon EC2, in turn reducing the monetary cost.

These computational time may depend on choices of frameworks and skills of implementations. However we think it is safe to say that the GraphNVP is significantly faster than other models in sampling for several reasons: the GraphNVP decoder does not involve additional chemical validity check (Jin et al., 2018), or grammatical validity-assurance for sampling (Kusner et al., 2017). Deterministic decoding of graphNVP further reduces generation time in practical scenarios since a latent vector is not needed to be decoded multiple times as done for JT-VAE.

### 4.3 SMOOTHNESS OF THE LEARNED LATENT SPACE

Next, we qualitatively examine the learned latent space $z$ by visualizing the latent points space. In this experiment, we randomly select a molecule from the training set and encode it into a latent vector $z_0$ using our proposed model. Then we choose two random axes which are orthogonal to each other. We decode latent points lying on a 2-dimensional grid spanned by those two axes and with $z_0$ as the origin. Figure 4 shows that the latent spaces learned from both QM9 (panel (a)) and ZINC dataset (panel (b)) vary smoothly such that neighboring latent points correspond to molecules with minor variations. This visualization indicates the smoothness of the learned latent space, similar to the results of existing VAE-based models (e.g., (Liu et al., 2018; Ma et al., 2018)). However, it should be noted that we decode each latent point only once unlike VAE-based models. For example, GVAE (Kusner et al., 2017) decodes each latent point 1000 times and selects the most common molecule as the representative molecule for that point. Because our decoding step is deterministic such a time-consuming measure is not needed. In practice, smoothness of the latent space is crucial for *decorating* a molecule: generating a slightly-modified graph by perturbing the latent representation of the source molecular graph.

### 4.4 PROPERTY-TARGETED MOLECULE OPTIMIZATION

Our last task is to find molecules similar to a given molecule, but possessing a better chemical property. This task is known as *molecular optimization* in the field of chemo-informatics. We train a linear regressor on the latent space of molecules with quantitative estimate of drug-likeness (QED) of each molecule as the target chemical property. QED score quantifies how likely a molecule is to be a potential drug. We interpolate the latent vector of a randomly selected molecule along the direction of

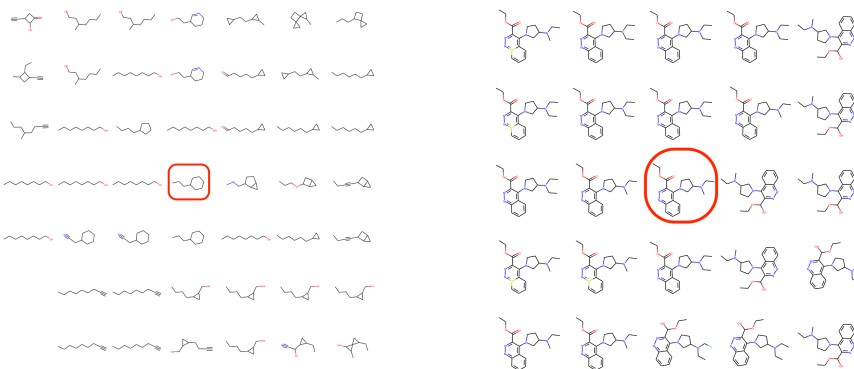

Figure 4: Visualization of the learned latent spaces along two randomly selected orthogonal axes. The red circled molecules are centers of the visualizations (not the origin of the latent spaces). An empty space in the grid indicates that an invalid molecule is generated. Left: Learned latent space for QM9. Right: Learned latent space for ZINC.

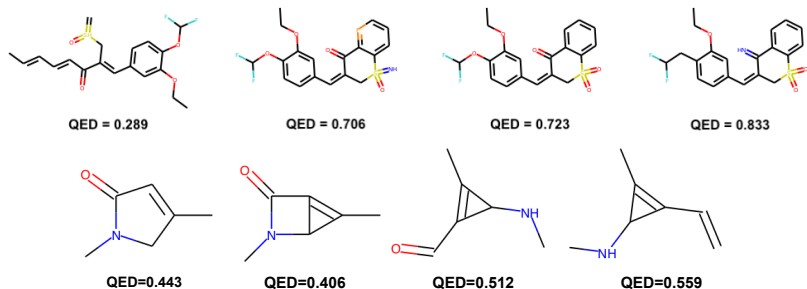

Figure 5: Chemical property optimization. Given the left-most molecule, we interpolate its latent vector along the direction which maximizes its QED property. Upper: Molecule optimization for ZINC. Lower: Molecule optimization for QM9.

increasing QED score as learned by linear regression. Figure 5 demonstrates the learned latent space and a simple linear regression yields successful molecular optimization. Here, we select a molecule with a low QED score and visualize its neighborhood. However, we note that the number of valid molecules that can be generated along a given direction varies depending on the query molecule. We show another property optimization example on QM9 dataset in the appendix.

Although we could perform molecular optimization with linear regression, we believe an extensive Bayesian optimization (e.g., (Jin et al., 2018; Kusner et al., 2017)) on the latent space may provide better results.

## 5 CONCLUSION

In this paper, we proposed GraphNVP, an invertible flow-based model for generating molecular graphs. Specifically, the proposed model is the first fully invertible model for the whole graph components: both of node attributes and an adjacency tensor are converted into latent vectors through two novel invertible flows. Our model can generate valid molecules with a high uniqueness score and guaranteed reconstruction ability with very simple invertible coupling flow layers. The proposed model enjoys a fast graph generation; faster in order of magnitude than other graph generation models in our implementation. In addition, we demonstrate that the learned latent space can be used to search for molecules similar to a given molecule, which maximize a desired chemical property.

## 5.1 Open Problems

As the first paper for fully invertible graph generation models, we identified several open problems of this research direction. One is the permutation-invariant graph generation, which is essentially difficult to achieve by coupling-based flow layers. Another is the number of nodes in generated graphs. The current formulation of the GraphNVP must choose the maximum number of nodes within generated graphs. This is the limitation of one-shot generative models compared to iterative ones. Incorporating external validity checks would improve the validity of the generative model. There is a possibility that overfitting causes the lower validity and novelty. If this is the case then it is interesting to devise a good regularizer for reliable graph generations. Additionally, we believe more exploration of the reasons contributing to the high uniqueness ratio of the proposed model will contribute to the understanding of graph generation models in general.

We will provide our implementation of the proposed GraphNVP in near future.

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

## A    NETWORK ARCHITECTURE DETAILS

For QM9 dataset, we use a total of 27 adjacency coupling and 36 node feature coupling layers. For ZINC dataset, we keep the number of coupling layers equal to the maximum number of atoms a ZINC molecule can have, 38. We model affine transformation (both scale and translation) of an adjacency coupling layer with a multi-layer perceptron (MLP). As mentioned in the main text, we utilize both node assignments and adjacency information in defining node feature coupling layers. However, we found affine transformations can become unstable when used to update the feature matrix with Relational-GCN (RelGCN). Therefore, we use only additive transformations in node feature coupling layers.

We initialize the last layer of each RelGCN and MLP with zeros, such that each affine transformation initially performs an identity function.

We train the models using Adam optimizer with default parameters ($\alpha = 0.001$) and minibatch sizes 256 and 128 for QM9 and ZINC datasets. We use batch normalization in both types of coupling layers.

## B    TRAINING DETAILS

For training data splits, we used the same train/test dataset splits used in (Kusner et al., 2017). We train each model for 200 epochs. We did not employ early-stopping in the experiments. We chose the model snapshot of the last (200) epoch for evaluations and demonstrations. All models are implemented using Chainer-Chemistry[1] and RDKit[2] libraries.

## C    EFFECT OF TEMPERATURE

Following previous work on likelihood-based generative models (Kingma & Dhariwal, 2018), we sampled latent vectors from a temperature-truncated normal distribution. Temperature parameter handles uniqueness and validity trade off. Sampling with a lower temperature results in higher number of valid molecules at the cost of uniqueness among them. How temperature effects validity, uniqueness, and novelty of generated molecules is shown in Figure 6. Users may tune this parameter depending on the application and its goal. In our experiments we chose 0.85 and 0.75 as the temperature values for QM9 and ZINC models respectively.

## D    EFFECT OF ADJACENCY TENSOR IN GRAPHNVP COUPLING

We performed additional experiment to quantify the effect of $A$ introduced in the node feature coupling. We trained an ablation model, which replace the RelGCN layer with an MLP which does not use $A$. For QM9 dataset the validity drops to $41.8 \pm 1.26\%$, about half the validity of original GraphNVP model.

---

[1]https://github.com/pfnet-research/chainer-chemistry
[2]https://github.com/rdkit/rdkit

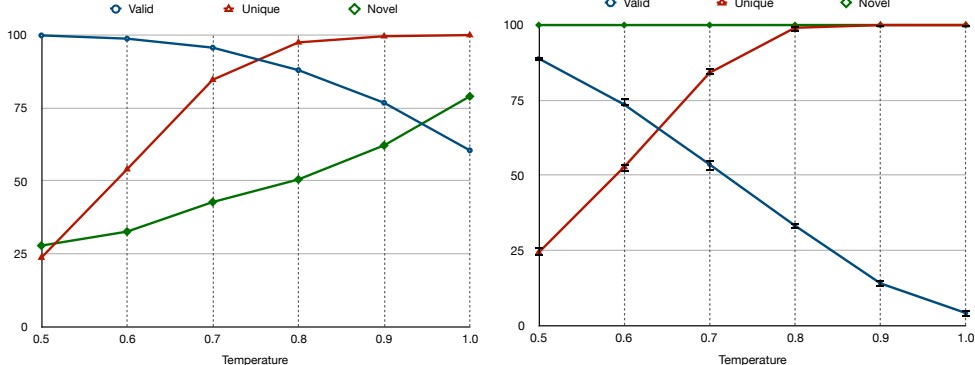

Figure 6: Impact of temperature on the quality of graph generation. Sampling with a smaller temperature yields more valid molecules but with less diversity (uniqueness) among them. Each experiment is performed five times and the average is reported in this figure. Left panel: impact of temperature on sampling from latent space of QM9. Right panel:Impact of temperature on sampling from latent space of ZINC.

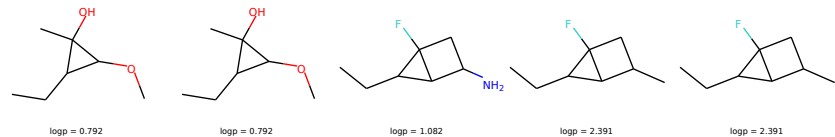

Figure 7: Chemical property optimization. We select a molecule from QM9 dataset randomly and then interpolate its latent vector along the axis which maximizes water-octanol partition coefficient (logP).

## E  ADDITIONAL VISUALIZATIONS

Fig. 7 illustrates an example of chemical property optimization for water-octanol partition coefficient (logP) on QM9 dataset.

