# OpenReview forum: "GraphNVP: an Invertible Flow-based Model for Generating Molecular Graphs"
_ICLR.cc/2020/Conference — Reject_

### Official Review · AnonReviewer1 · 2019-10-25
**Official Blind Review #1**

**Rating:** 3

**Review:**

Contributions:
1. This paper proposes an invertible flow-based method for the one-shot graph generation.
2. The paper demonstrates their method on a molecular graph generation task.
3. Empirical results show the effectiveness of the proposed method.

The merit of the proposed invertible flow method is two folds. First, it can guarantee a one hundred percent reconstruction accuracy. Second, it can be adapted to generate graphs with various types (such as molecules) without incorporating much domain knowledge. Below are my concerns regarding this paper.

[Page 7, Table 2] My first concern is: does the reconstruction performance matters in the graph generation case? Typically a lower reconstruction error does not mean a worse model to generate reasonable new graphs. So the reconstruction error should accompany with other criterions. In Table 2, I can see CD-VAE and JT-VAE does better in generating valid, novel and unique graphs. So I wonder whether it worth sacrificing novelty to pursue a perfect reconstruction.

[Page 7, Sec 4.2] The authors mention they cannot reproduce the decoder of CG-VAE and JT-VAE. So I expect they mention somewhere in this paper that they will release their code once published.

[Page 5, Sec 3.3.1] The authors should be explicit by saying we replace sliced matrices (z_X[l^-,:,] in Eq. 2 and z_A[l^-,:,:] in Eq. 4) with masked matrices rather than just saying "Eqs. (2,4) are implemented with masking patterns".

[Page 5, Sec 3.3.1] Can you explain the gain of masking? To my understanding, even with masking you still need a sequence of N coupling layers to update each node once.

[Page 5, Sec 3.3.1] The second paragraph in Sec 3.3.1 is confusing to me. The masking scheme indeed makes the whole process, not permutation invariant. But I'm confusing about the way you fix it. Can you explain your "permutation invariant coupling"? E.g., why you need to change the indexing on the non-node axis?

Overall, I think the method proposed in this paper sacrifices some more import aspects in graph generation such as novelty and uniqueness by introducing an invertible flow architecture. And some parts in the paper may require a significant re-writing, such as Sec 3.3.1.

**Experience Assessment:**

I have read many papers in this area.

**Review Assessment: Checking Correctness Of Derivations And Theory:**

I carefully checked the derivations and theory.

**Review Assessment: Checking Correctness Of Experiments:**

I carefully checked the experiments.

**Review Assessment: Thoroughness In Paper Reading:**

I read the paper thoroughly.

---

> ### Author Response · Authors · 2019-11-13
> **Reply to official blind review #1**
>
> Thank you for your review!
>
> Reconstruction rate:
> We think the 100% reconstruction rate is very important in lead optimization application of drug discovery domain. Please refer to the reply to reviewer #2.
>
> Code release:
> We consider to release the implementation of the proposed GraphNVP and wrote so in the updated manuscript.
>
> Slicing and Masking:
> [l^-, :, :], slicing operation is implemented as multiplication with a mask as shown in Figure 2.
>
> The effect of masking:
> Masking is used to update only a part of the input leaving the rest unchanged. This guarantees that the affine transformation is reversible. We started from trying maskings such that allows multiple nodes are updated simultaneously, but they never work fine. So we think the main effect of the one-node-at-one-layer masking is to focus the representation power of relatively simple affine couplings to one node.
> Yes, we need at least N layers for the graph of N nodes. This may sound huge, but in invertible image generation models such as glow typically requires 100~ and more layers. In addition, each affine layer is so simple that computational cost is quite small, so many layers does not harm training and sampling so much.
>
> Permutation-invariant coupling:
> With ```"`permutation” we mean the invariance concerning the order of nodes. If we want to make the couplings invariant to the order of nodes, we cannot use the coupling (masks) like in Eqs (2.4) and Fig. 2 where the layer index l also specifies the node to be computed. Thus remaining axes to sweep (with layer index l) is the non-node axes in X and A.
> Unfortunately this attempt did not work well, possibly because of the sparsity in atoms (most atoms are Carbon) and the bond types (most bonds are ``virtual (not connected’’). If we mask the most frequent atom/bond type, then the remaining tensors lose its information.
>
> Concerning the performance:
> In graph generation, “the number of valid, unique, and novel molecular graphs” within a unit time is a practically important metric. The validity of the graphNVP samples are relatively low, but still keeps 40%. Uniqueness and novelty are comparable with the best existing models. In contrast, sampling time is x30~x100 times shorter. Thus we can simply obtain more (x10 ~ x40 times) valid and unique molecules in the same computation time. In this sense, the proposed GraphNVP is not clearly inferior to other (stronger) baselines.
> Please find the updated Sec. 4.2.1 and our answer A4 for reviewer #3.

---

### Official Review · AnonReviewer2 · 2019-10-25
**Official Blind Review #2**

**Rating:** 3

**Review:**

This paper presents a new reversible flow-based graph generative model wherein the whole graph i.e., representative attributes such as node features and adjacency tensor is modeled using seperate streams of invertible flow model. This allows training of generative model using exact likelihood maximization over the underlying graph dataset.The model avoids encoding any domain specific heuristics and thus can be applied to any structured graph data. The paper focusses it applicability for molecular graphs. Given that this approach avoids sequential generation of graph, it is faster by an order of magnitude than prior models for molecular generation. Empirical experiments on couple of molecular graph data suggets that GraphNVP approach performs as well as prior approach but albeit without any rule checker.

My major concern with such invertible models is "scalability". Given that flow-based model are required to retain the original dimension its applicability is limited to low dimensional feature vectors. In the case of GraphNVP, this means limited number of node labels as well as edge labels. Additionally, since it limits adjacency tensor, this would lead to modeling graphs with few nodes. However, if integrated with encoder-decder model some of these limitation can be overcome. Given this major weakness and with limited novelty (i.e., extending to adjacency tensor), I am inclined to reject this paper. I shall improve my rating if GraphNVP is applied to general graph structures - synthetic / real.

Few more limitations:
1. Although paper claims one-shot generation of graphs, in reality it seems otherwise. Since every layer processes only on single node, overall it operates sequentially from one node to another.
2. Moreover, this same sequential processing yet again limits it applicability to small graphs.
3. As in MolGAN, the direct generation of adjacency tensor leads to training with fixed size graphs i.e., through the addition of virtual nodes.  It is not possible to train model with variable number of nodes.
4. As pointed by authors, their model is not node permutation invariant.

Clarification:
1. Are the function 's' and 't' fixed across time ? For QM9 with max of 9 atoms and 27 layers, each atom attribute is processed multiple times. Are they processed using same functionality of s and t ?
2. Is it possible to model permutation invariance by augmenting the training data using multiple permutation of nodes such as BFS, DFS, degree, k-node (see GRAN) ?
3. I understand GraphNVP can reconstruct perfectly. But I fail to note the actual significance of such metric. If it reconstruct 100% or not how does it matter ? What matters is unniqueness, validity and novelty.
4. Can you please compare inference time ?
5. How difficult is it to integrate validity checker with your generation process ? Can we have some comparison using it ?

Minor:
1. In eq (2) please use different notation for layer 'l' and node 'l'.
2. Page 4, penultimate line: So as functions s and t -? To model functions s and t

**Experience Assessment:**

I have read many papers in this area.

**Review Assessment: Checking Correctness Of Derivations And Theory:**

I carefully checked the derivations and theory.

**Review Assessment: Checking Correctness Of Experiments:**

I carefully checked the experiments.

**Review Assessment: Thoroughness In Paper Reading:**

I read the paper thoroughly.

---

> ### Author Response · Authors · 2019-11-13
> **Reply to the official blind review #2**
>
> Thank you for your review!
>
> Scalability:
> We primarily conduct this research for molecular graph applications. And many researches (including enc-dec (VAE)-based models) only focus on small molecular graphs, whose size of atom vocabulary (node labels) and a number of atoms (graph sizes) are pretty limited. Despite the limited graph sizes, the solution space of molecular graphs is exponentially large ( e.g. 50-atom molecule with 10 possible atom labels has 10^50 possible configurations) so generating valid small molecules is not a trivial problem.
> We do not intend to claim that the GraphVAE is suitable for large molecules and graphs in other domains such as social networks.
>
> At the same time, we thank you for pointing out the inherent limitations of the proposed invertible model, such as the permutation-invariance and variable-size graph generation. We believe the most important contribution of this work is to propose a fully invertible graph generation formulation, first in the literature.
> We also think it is very important to identify properties and limitations of invertible graph generation models, and share readers the technical problems of fully invertible graph generation models, opened to the community by our research. We augmented the updated manuscript with the above discussion. To present the latter point more clearly, we added the “open problems” subsection at the conclusion section.
>
> Concerning one-shot and iterative generation:
> it is a good question. The generation process of the GraphNVP seems apparently sequential, but in fact the sequential process is perfectly deterministic. The final graph structure output is completely determined when the latent z vectors are sampled because of invertibility. Iterative generation methods in Table 1, on the other hand, perform some disambiguation operations in each sequential step: e.g. computing scores and perform pruning to choose a node of focus, and stochastic sampling of edge types, and so on. There is a huge gap between these, so we believe it is not appropriate to call the GraphNVP as an iterative generator.
>
> The scale and translation networks (s, t);
> these networks are time-variant. Each layer has its own s() and t(). If multiple coupling layers are applied to a specific node, then each layer has different parameters.
>
> Data augmentation:
> We have not considered augmenting the training data graphs to make it permutation-invariant. We are not bright at these techniques, so rather we want to ask for experts whether these techniques can be a remedy. Although we are not sure about permutation-invariance, suggested augmentation seems quite beneficial to virtually increasing the training data to prevent overfitting. Thank you so much for your idea.
>
> Reconstruction rate:
> We think the high reconstruction ratio means the learned model can capture any minute changes in the graph. This is a basic requirement for graph models. Also, reconstruction in drug discovery domain is important when we consider lead optimization. Assume we already have “seed” molecule (drug) which has a nice property, and we would like to generate many “similar” molecules. 100% reconstruction rate means that we have a theoretical guarantee that we can obtain latent variable Z of this seed molecule, so that we can search the neighborhood of this Z to generate similar molecules. If reconstruction accuracy is low, decoding a single latent vector multiple times will result in different molecules. Previous work sidestep this issue by decoding a vector several times and choosing the most common generated molecule as the output.
>
> Importance of uniqueness, validity, and novelty:
> In performance evaluation of graph generative models, and it is important to assess a combination of uniqueness, validity, novelty, AND the computational speed. The graphNVP can sample molecular graphs x30-x100 times faster. Validity is somewhat low, but the uniqueness and the novelty are comparable with best models.
> So the graphNVP can generate more valid, unique, and novel graphs compared to the counterparts within the same computational time.
> Please find the Sec 4.2.1 for revised explanations, and our answer A4 for reviewer #3.
>
> Inference time:
> We are not clear what do you mean by “inference” time. If you mean the computational cost for the forward path (encode), it is the same with the backward path (sampling) for our simple affine couplings. If you mean the “time for training/parameter optimization”, for example, it takes 10 hours for 200 training epochs for QM9 dataset with a single commercial GPU, no parallelism.
> We think it is not so much difficult to incorporate the validity checker. But if it costs too many CPU times, it loses the main practical advantage of the proposed method: sampling speed.

---

### Official Review · AnonReviewer3 · 2019-10-25
**Official Blind Review #3**

**Rating:** 3

**Review:**

In this paper, a GraphNVP framework for molecular graph generation is proposed. The main difference from the previously proposed models is the use of the invertible normalizing flow idea for the generative model, which doesn’t require a separate decoder for sampling. This architecture is implemented with coupling layers combined with a multi-layer perceptron. The model is evaluated and compared on QM9 and ZINC chemical molecular datasets.

This method combines a number of existing techniques to obtain a new model for the molecular graph generation problem. The paper is very well written.

I have several concerns with regards to this model and the proposed algorithm:
1. How many parameters does GraphNVP model have? The coupling layers should have at least O(LN^2R) and that must be multiplied by the number of MLP parameters of the adjacency tensor which I suppose is of order O(N^2R), is this correct? This number must be huge. How can one ensure that such a model does not overfit? Moreover, 100% reconstruction accuracy is rather an indicator that it actually does overfit, isn’t it?
2. I’m concerned whether the use of dequantization for this particular application is valid. Indeed, images are usually modeled as vectors taking values in [0, 255] and adding a uniform on [0,1] variable actually corresponds to noise. However, the graph adjacency tensor takes either 0 or 1 values and adding a similar uniform variable (potentially scaled by 0.9) is actually more than simply adding noise. Say one value is 0 and added 0.8, while the other is 1 and added 0.1; the transformed variables are now much closer to each other. Therefore, the likelihood of the transformed variables is going to be significantly different from the original one. Although one can indeed recover the original graph from the dequantized one, I doubt that there is a correspondence between two likelihoods extrema. This also contradicts one of the motivations to this paper that this approach uses precise log-likelihood. Could you please comment on this?
3. I am confused by this sentence: “Our objective is maximizing the log likelihood (Eq. 1) of z over minibatches of training data.” Does this mean that log(p_z(z)) is the objective? If yes, how does this relate to maximum likelihood?
4. Given the high cost of wet-lab experiments, the runtime of a method for the drug discovery application is much less important than the quality of the obtained results. Is it actually more important to have a sampling time decrease from 100/400s to 4s or does the higher quality of results should matter more? If the latter, are there any other advantages of GraphNVP over other models?

**Experience Assessment:**

I do not know much about this area.

**Review Assessment: Checking Correctness Of Derivations And Theory:**

I carefully checked the derivations and theory.

**Review Assessment: Checking Correctness Of Experiments:**

I assessed the sensibility of the experiments.

**Review Assessment: Thoroughness In Paper Reading:**

I read the paper thoroughly.

---

> ### Author Response · Authors · 2019-11-13
> **Reply to the official blind review #3**
>
> Thank you very much for your review!
>
> A1:
> In all invertible models including existing image generation models such as GLOW, the number of parameters is quite huge. However, computations of each coupling layer are kept simple (for invertibility requirements) so the model’s representation power is regularized even though the number of parameters is huge as you pointed out.
> Also, the maximum number of atoms is 9 for QM9 dataset and 28 for Zinc dataset respectively, so N^2 part is not too big in our target domain. Of course, reducing the number of parameters is important in some applications, but is out of the scope of our study.
>
> Thank you for reminding an important point, overfitting. Yes, in our case, actually the overfitting happens and causes a problem. But let us make clear that 100% reconstruction and overfitting are different and unrelated issues.
> 100% reconstruction accuracy: it is achieved not as a result of overfitting. The affine transformations used in the coupling layers are designed to be invertible, such that running the reverse step of coupling layers on an output can reconstruct the input (dequantized graph components) exactly.
> Overfitting: In our case, the overfitting will make the generated molecules from the samples of p(z) mostly the training molecules, which should be evaluated by “Novelty” value in Table 2. Because we initialize the last layer of each RelGCN and MLP with just identity functions, it will never generate any valid molecule including the training molecules. Thus the low “Novelty” suggests the overfitting as the learning proceeds, which also means, in contrast, that we may be able to prevent the overfitting by monitoring the “Novelty” value during the course of training. Also, it may be good to add some regularization terms to prevent the overfitting. So we are optimistic to improve our model although the overfitting happens to our model as you suggested.
> On the contrary, in the application, it is important to generate valid and novel molecules efficiently. Because our model can generate unique molecules in high probability, the ratio of the generating novel molecules, V(alidity) x U(niqueness) x N(ovelty), will be high enough compared with the existing methods except for the JT-VAE and CG-VAE which requires complicated decoders. Please find A.4 for additional explanations.
>
>
> A2:
> We think this dequantization cause no problems in the set up specified in the paper. The point is that the range of dequantized 1-entries in A and the range of dequantized 0-entries in A does not overlap. Setting 0 < c < 1.0 ensures this condition in our set up.  As long as the above condition is met, the actual value of dequantized entries should not be important. During the training, the model is trained to correctly recover dequantized tensors with c=0.9 magnitude noise addition. Therefore we expect the noises on dequantized tensors are also not fatal in the test (generation) phase.
> Please also note we compute the likelihood of an adjacency tensor in the space of latent variables Z_A, not in the space of dequantized A. So, the difference in the dequantized A values do not directly correspond to the difference in the likelihood. An entry of dequantized A with value = 0.8 is not necessarily close to a dequantized entry with value=1.0 in the z_A space.
>
> A3:
> Excuse us for confusing descriptions. Our objective is the log p_{G} (G), the l.h.s. of Eq. 1. We will modify the sentence according to your comment.
>
> A4:
> Computational (sampling) time is still important. It affects “the number of valid and unique molecular graphs” we can collect within a unit time. The validity of the GraphNVP samples are relatively low, but still keeps 40%. In contrast, the sampling time is x30~x100 times shorter. Thus we can simply obtain more (x10 ~ x40 times) valid and unique molecules in the same computation time.
> Once we obtained the generated molecule, we usually calculate/predict the value of specific property in computer to check whether the molecule has a desired value. Thus generating many molecules increases the chance to discover molecule with required property. Assume we need to prepare 1 million unique & valid molecules from ZINC training set.  With a very rough estimate, we expect the graphNVP requires 1.1 hours to generate 1M molecules, while JT-VAE requires 33 hours (1.5 days) and GVAE takes 121 hours (5 days). Such slow graph generations would harm the productivity of the R&D projects.
> Further, this will reduce the usage of cloud computing servers such as Amazon EC2, in turn reducing the monetary cost.
> We incorporated this issue to Sec. 4.2.1 of the updated manuscript.

---

### Decision · Program_Chairs · 2019-12-19

**Decision:**

Reject

**Comment:**

The authors propose an invertible flow-based model for molecular graph generation. The reviewers like the idea but have several concerns: in particular, overfitting in the model, need for more experiments and missing related work. It is important for authors to address them in a future submission